# Development of Simplified Seedling Transplanting Device for Supporting Efficient Production of Vegetable Raw Materials

Luhua Han [1,*] , Daqian Xiang [1], Qianqian Xu [1], Xuewu Du [1], Guoxin Ma [2] and Hanping Mao [1,*]

[1] Key Laboratory of Modern Agriculture Equipment and Technology, Ministry of Education, Jiangsu University, Zhenjiang 212013, China; 2212116080@stmail.ujs.edu.cn (D.X.); 2222216054@stmail.ujs.edu.cn (Q.X.); 2222216031@stmail.ujs.edu.cn (X.D.)

[2] High-Tech Key Laboratory of Agricultural Equipment and Intelligence of Jiangsu Province, Zhenjiang 212013, China; mgx@ujs.edu.cn

[*] Correspondence: hanlh@ujs.edu.cn (L.H.); maohp@ujs.edu.cn (H.M.)

**Featured Application: Automatic seedling transplanting machinery.**

**Abstract:** Efficient greenhouse production has a great supporting role in the development of vegetable agricultural and sideline product processing. In this paper, a simplified automatic transplanting device was designed and evaluated in a laboratory. The device mainly consists of a seedling pick-up gripper, a transplanting manipulator, two conveyors and a control system. The flexible multi-pin gripper was designed to effectively grasp, hold, and release seedlings. Through a combination innovation of the linear modules, the transplanting manipulator was designed to move the seedling gripper to the desired working position. The conveyors were the pallet-type double-row chain transmission system for automatic feeding of plug trays and growth pots. The control system was developed to coordinate each of the aforementioned function units. The multi-factor orthogonal and transplanting performance experiments were carried out under the standard seedling agronomy. The results showed that the transplanting frequency and the pick-up depth significantly affected the transplanting quality. When the transplanting frequency was 15 plants/min, the tightened spring force was 1.2 N, and the pick-up penetration depth was 35 mm, the optimum effects of automatic transplanting seedlings could be achieved. The maximum success in transplanting seedlings was 95.47% for local vegetable crops. The developed prototype could realize less waste of seedling resources at the farm level.

**Keywords:** greenhouse production; plug seedling; automatic transplanting; flexible grasping; orthogonal experiment

## 1. Introduction

With the economic development and the consumption upgrade, urban and rural residents have increasingly diversified food demands. Pickled vegetables are commonly flavored non-staple foods with a yield of 6.50 million tons annually in China [1,2]. These minimally processed foods, such as pickled cucumber chilli sauce, are fresh, crisp, hot, and sour while maintaining the same nutritional and organoleptic (sensory) quality of fresh vegetables [3]. So pickled vegetables are considered one of the important agricultural by-products of the Chinese diet, without which any meal is imperfect. In recent years, dehydrated vegetables have been increasingly widely used in convenience foods, desserts, breads, condiments, snack foods and other scenes with the progress of dehydration technology and the acceleration of residents' life pace [4]. Some supermarkets in large and medium-sized cities and remote areas also have a large demand for dehydrated vegetables. The use of processed methods to transform plant-derived foods into ready-to-eat products may differ in thousands of ways for various pickles and dehydrated vegetables. But their raw materials are mainly fresh vegetables. The efficient facility agriculture is the

basis to ensure the yield and marketability of stubble vegetables, which greatly supports the development of vegetable agriculture and sideline product processing [5]. So, it is significant to develop modern greenhouse agriculture, which can supply various nutritious foods regardless of seasons. Plug trays are actively used to grow seedlings, which can well adapt to further growth and development after transplantation. Their other advantages are uniform plant quality, effective production scheduling, and efficient materials handling. According to the statistics, 350 billion plants of professional plug seedlings are produced annually in China [6]. Before reaching the consumers, these seedlings will be transplanted to individual pots of growing flats for further growth and may be handled many times [7,8]. Manual transplanting in large-scale production might be labor-intensive and less uniform than mechanical operation [9,10]. It is also expensive and time-consuming. The mechanized transplanting technology has a high seedling preservation rate and a fast growth process, which can significantly contribute to agri-food waste mitigation at the farm level. The use of robots in the greenhouse has been suggested to carry out these repetitive and burdensome tasks in an accurate and reliable way [11]. The labour shortage has also made greenhouse producers seek help from mechanized efficient production of raw materials for these ready-to-eat products [12].

Much research on automated transplanting systems has been made worldwide, which wants to develop effective processing methods for seedling transplanting with less damage [13–16]. Relying on the advanced industrial robot technology, the developed countries in Europe and America began research on developing fully automatic transplanters for greenhouse plants several years ago. A robot was studied to transplant bedding plants with computer graphics and simulation [17]. The testing results showed that the robot could transplant most seedlings in the low damage condition at an average cycle time of 3.3 min for one 36-cell growing flat. Despite being inefficient, this study demonstrates that it is feasible to use robots for automatic transplanting. Many mechanical and horticultural factors were also checked to understand their influences on the transplanting quality [18]. These studies provide the design basis for further development of dedicated devices. An automatic robotic transplanting system was designed and evaluated for bedding plants, such as cucumber (*Cucumis sativus* L.), tomato (*Solamum lycopersicum* L.), and other varieties [7]. Further, some tests showed that the robotic transplanting performance was affected by the accuracy of different end-effector positioning and the seedling conditions. In the developed countries of facility horticulture, fully automatic transplanters have been promoted for efficient vegetable production, meaning less waste of seedling resources at the farm level.

The modern greenhouse transplanting systems are internationally developed to use the automatic production line mode. These foreign machines perform well, but they are expensive to manufacture and cumbersome to maintain. It may be difficult to be widely used by small growers [8]. In China, the small-scale gardening area, as the main production mode, accounts for 70% of the total horticultural area [19]. It is necessary to develop a small greenhouse transplanting machinery. Several research studies have been done on greenhouse transplanting production. Some key parts, such as the pick-up device, the end-effector, and so on, were designed to improve the level of seedling mechanized transplanting [20–25]. Based on this structure design, the machine vision system was also used to evaluate the seedling quality, allowing for automatic sorting and subsequent transplanting of tray seedlings [26]. The existing design concepts range from simple devices to custom equipment, and some design is with the capabilities of dexterous hands. There are many variations in the dimensions of plug trays and growing flats. Some commercially available robots can respond to these variations, requiring only changes in the software program or its input control values [27]. However, most existing designs have low-level flexible automation and are not adaptable to various seedling sizes and shapes. There is still a certain gap in the transplanting efficiency and success rates compared with the foreign equipment.

Efficient transplantation can ensure a high yield of fresh vegetables and further promote the customized processing of agricultural by-products. In this paper, a simplified automatic transplanting system was developed for high efficiency and low damage of seedling planting. The corresponding transplanting performance was evaluated under actual production conditions. This study would provide innovative ideas for the development of fully automatic transplanters to support the high-quality production of vegetable raw materials.

## 2. Materials and Methods

### 2.1. Design Conditions

As shown in Figure 1, the seedling growth and development are uniform as produced in plug trays. It is a key step for greenhouse production to timely and efficiently transplant seedlings from a growth tray into some larger pots. Some design conditions for seedling transplanting are established according to cultural practices for vegetable production in China.

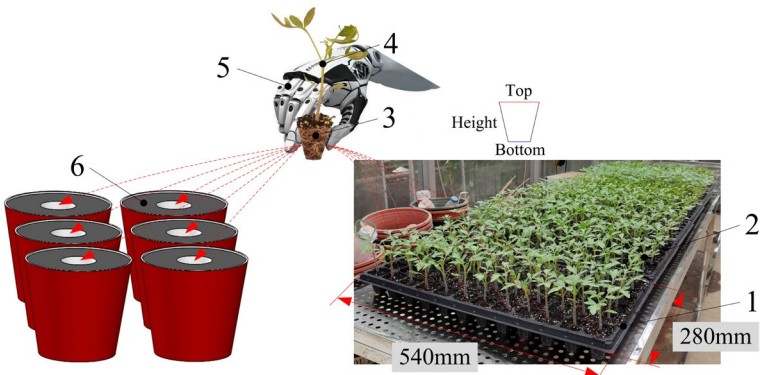

**Figure 1.** Greenhouse production of transplanting seedlings from growth trays to larger pots: 1. plug tray; 2. plug seedling; 3. root lump; 4. seedling plant; 5. transplanting gripper; 6. growth pot.

(1) The widely used soft plastic trays were made of injection molded polystyrene. The overall dimensions are 280 mm width × 540 mm length for each type of tray. Two widely used trays have inverted truncated pyramid-type cells with the interval arrangement of 16 × 8 (128-cell tray) and 12 × 6 (72-cell tray), respectively. The cell dimensions are 48 mm height × 32 mm top × 14 mm bottom and 45 mm height × 40 mm top × 20 mm bottom for the 128-cell tray and the 72-cell tray, respectively.

(2) The transplanting machine should adapt to different kinds of tray cells and irregular-shaped bedding plants by adjusting structural parameters. The limited growing space of the tray cell may make the seedling roots adhere around the perimeter of the tray cell. Therefore, it needs moderate mechanical action to grasp seedlings and extract them from tray cells.

(3) To ensure the survival rate, the flexible design is important to minimize bruising damages when picking up the seedling in transplanting. The transplanting machine needs to be equipped with a dexterous gripper that can grasp and plant a variety of seedlings. Its design also influences the allowable approaches and departures of the transplanting system from the work object.

### 2.2. Innovative Design of Transplanting Mechanism

During this stage of research, two parallel tasks have been identified as necessary to build a basis for the development of a successful transplanting system. They are the gripper development and the materials flow analysis. Figure 2a shows the mechanism schematic diagram of the designed robotic transplanting workcell. Its overall structure is a simple gantry structure with a robotic mechanism. The transplanting manipulator was designed with the X-Y Cartesian coordinate system. In particular, the seedling gripper is

installed on the slider 5. The belt pulleys 4, 6, and slider 5 constitute an X-axis single-drive translating system. The Y-axis linear motion of the seedling gripper is performed using the double-drive synchronous rising and falling systems with belt pulleys 1, 3, & slider 2, and belt pulleys 7, 9, & slider 8, respectively. So the transplanting manipulator moves the seedling gripper to approach a tray cell and grasp the seedling (Figure 2a: working trajectory ①), transfer the seedling along a stable trajectory from the growth tray to the large pot (Figure 2a: working trajectory ②), plant the seedling into the growth pot (Figure 2a: working trajectory ③), and finally return for the next transplanting operation (Figure 2a: working trajectory ④).

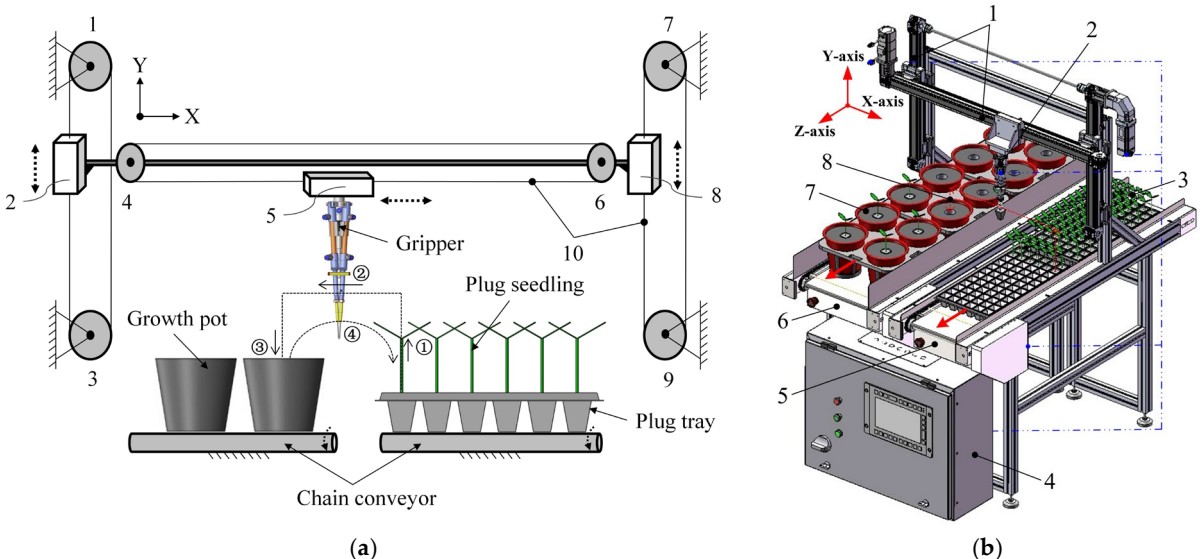

(**a**)                    (**b**)

**Figure 2.** Structure drawing of the simplified greenhouse robotic transplanting workcell: (**a**) mechanism principle: 1, 3, 4, 6, 7, 9. belt pulley; 2, 5, 8. slider; 10. synchronous belt; (**b**) mechanical structure: 1. transplanting manipulator; 2. seedling gripper; 3. plug seedling; 4. control system; 5. chain conveyor for plug trays; 6. chain conveyor for growth pots; 7. growth pot; 8: working trajectory.

Figure 2b shows the mechanical structure drawing of the designed robotic transplanting workcell. The seedling gripper was designed as a flexible multi-pin grasping type working unit, which could pick up the seedlings with low damage [4]. The precise synchronous belt linear actuators were used to construct the H-shaped transplanting mechanical manipulator with X-Y two-dimensional motion. Each direction of motion was driven by a servo motor system, which could achieve the precise positioning of the gripper. Two double-row chain-type conveyors were configured in parallel for continuously moving these plug seedlings and growth pots to the working position of the seedling gripper. A set of multi-axis motion controls was developed to automate the seedling transplanting operation.

### 2.2.1. Seedling Gripper

The successful integration of a robot with seedling transplanting requires an operational pick-up gripper that can grasp and hold a seedling. Figure 3a shows the mechanical structure drawing of the flexible pneumatic gripper using four elastic pick-up pins pulled by the cylinder fingers. It mainly consists of a cross-fixed bracket, several cylinder & rubber fingers, a tightening spring, a limit gasket, some support parts, and other pneumatic parts (quick joints, solenoid valves, magnetic switches, and so on). In the design, four-cylinder fingers of retractable pick-up pins were symmetrically allocated in the crossing form onto the rubber finger that was opening and closing with inner support. One end of each cylinder finger was hinged to the fixed bracket, and another end of the piston rod was consolidated to the pick-up pins threaded with an extension rod. So, the picking pins could

be pushed out to penetrate the seedling's root lump and pulled back under the tractive action of the double-acting cylinder. The tightening spring was looped over the cylinder fingers, which could apply a certain pre-tightening force to the retractable pick-up pins. This way, the picking pins could grasp and hold the root lump from four corners. The limit gasket in different sizes was used to modulate the opening degrees of the cylinder fingers. The gripper that was pneumatically driven under the control of some solenoid valves could effectively grasp, hold, and release plug seedlings.

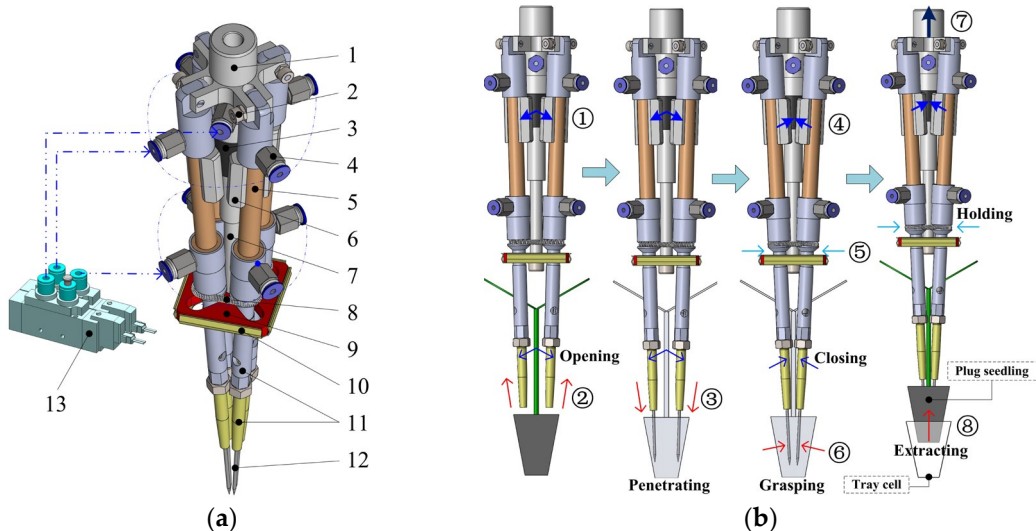

**Figure 3.** Structure drawing and operation schematic of the multi-pin grasping type seedling gripper: (**a**) overall mechanical structure: 1. fixed bracket; 2. revolute joint; 3. rubber finger; 4. pneumatic connector; 5. cylinder finger; 6. support plate; 7. support bar; 8: tightening spring; 9. support slot; 10. limit gasket; 11. extension rod; 12. pick-up pin; 13. pneumatic solenoid valve; (**b**) grasping operation: ① air inflating; ② pin withdrawing; ③ pin penetrating; ④ air deflating; ⑤ spring tightening; ⑥ pin grasping; ⑦ gripper moving; ⑧ pin extracting.

The operation of the pneumatic gripper for transplanting seedlings was described as follows. When preparing to pick up a seedling, the rubber finger is inflated (Figure 3b①). Meanwhile, the cylinder fingers withdraw multiple picking pins (Figure 3b②). So, the gripper is in an open state to approach the seedling. When picking up the seedling, the four-cylinder fingers push out their pins to penetrate the root lump of the seedling at a certain depth along the tray cell's wall (Figure 3b③). Then, the rubber bag is forced to deflate and contract (Figure 3b④), and the four-cylinder fingers close due to the spring tension (Figure 3b⑤). Under the action of the tightening spring, the pick-up pins of four-cylinder fingers can grasp and hold the root lump (Figure 3b⑥). Finally, the end-effector moves up (Figure 3b⑦), which pulls out the seedling from its growth tray cell (Figure 3b⑧). When the seedling is released, the gripper is opened to loosen the tightly-held root lump. Then, the pick-up pins are retracted. So, the extracted seedling naturally falls for planting.

Figure 4 illustrates the schematic diagram of the grasping action of the gripper. To successfully grasp the seedling's root lump, the pick-up pins need to penetrate the root-soil as deeply as possible within the depth of the cell [8,14]. Here, it was assumed that the four pins of the seedling gripper were symmetrically distributed, and the root lump was an equilateral square frustum. According to the geometric relations of grasping action between the pick-up pins and the tray cell, a set of equations can be easily determined as follows.

$$\begin{cases} L_{CC'} = b - 2\Delta d_1 \\ L_{DD'} = a - 2\Delta d_2 \\ L_{CF'} = \sqrt{2}(b - 2\Delta d_1) \\ L_{DG'} = \sqrt{2}(a - 2\Delta d_2) \end{cases} \tag{1}$$

where $L_{CC'}$, $L_{DD'}$, $L_{CF'}$, and $L_{DG'}$ are the corresponding pick-up parameter dimensions, respectively, mm; $a$ and $b$ are the lower side length and the upper side length, respectively, mm; $\Delta d_1$ and $\Delta d_2$ are the side distances from the upper and lower pick-up pin inlets to the hole, respectively, mm.

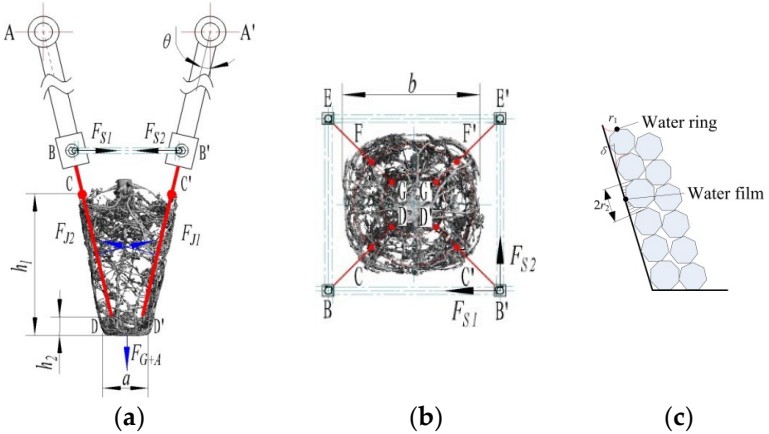

**Figure 4.** Schematic drawing of grasping action on the root lump of plug seedlings: (**a**) front view; (**b**) vertical view; (**c**) water film attraction.

The upright inclination angle of the pick-up pin along the grasping direction was set as $\theta$. Then, the penetration depth can be theoretically determined as follows.

$$L_{CD} = \frac{h_1 - h_2}{cos\theta} \tag{2}$$

where $L_{CD}$ is the corresponding pick-up penetration depth, mm; $h_1$ is the height of the root lump, mm; $h_2$ is the vertical distance from the end of the pick-up pin to the hole bottom, mm; $\theta$ is the pick-up upright inclination angle, °.

In this study, it was assumed that the root lumps of plug seedlings were of homogeneous body. The grasping forces were equal in all directions, and the pick-up pins were applied at the force action within the rigid range. Then, the effective equilibrium condition in the clamping and extraction directions can be expressed as follows.

$$\begin{cases} F_{J1} = F_{J2} \\ F_{S1} = F_{S2} \\ F_{J1} \times cos\theta = \sqrt{2}F_{S1} \\ 2(F_{J1} + F_{J2}) \times sin\theta \geq F_{G+A} \end{cases} \tag{3}$$

where $F_{J1}$ and $F_{J2}$ are the holding forces imposed on the pick-up pins, respectively, N; $F_{S1}$ and $F_{S2}$ are the grasping forces imposed by the bundling of the tightening spring, respectively, N; $\theta$ is the pick-up upright inclination angle, °; $F_{G+A}$ is the resultant force from the seedling's gravity and its adhesion force in the vertical direction, N.

The equivalent transformation was performed on the Equation (3). To successfully pull out the seedling plug, the tightening force of the spring on the pick-up pins should meet the following constraint condition.

$$F_{S1} \geq \frac{\sqrt{2}cos\theta \times F_{G+A}}{8sin\theta} \tag{4}$$

where $F_{S1}$ is the grasping forces imposed by the bundling of the tightening spring, mm; $\theta$ is the pick-up upright inclination angle, °; $F_{G+A}$ is the resultant force from the seedling's gravity and its adhesion force in the vertical direction, N.

Since the root lump of the plug seedling is the root-soil complex, a water film attraction has been formed between the root lump and its cell wall at some water contents [28]. The corresponding force values could be calculated as follows.

$$\begin{cases} F_{G+A} = F_G + F_{A1} + F_{A2} + F_{A3} \\ F_{A1} = 2k_1\pi r_1(\gamma_{\delta1} + \gamma_{\delta2} - \gamma_{\delta1|\delta2}) \\ F_{A2} = \sum 4\pi r_1\gamma_{LV}\cos\delta \\ F_{A3} = \sum \frac{3k_2\eta\pi r_2^4}{4t} \end{cases} \tag{5}$$

where $F_{G+A}$ is the resultant force from the seedling weight and its adhesion force in the vertical direction, N; $F_G$ is the seedling's gravity, N; $F_{A1}$, $F_{A2}$ and $F_{A3}$ are the equivalent water molecular attraction force, the equivalent water ring adhesion force and the equivalent water film adhesion force, respectively, N; $\gamma_{\delta1}$, $\gamma_{\delta2}$ and $\gamma_{\delta1|\delta2}$ are the surface tension of seedling substrates, the surface tension of the tray cell and their interfacial tension, respectively, N/m; $\gamma_{LV}$ is the surface tension of water, N/m; $\delta$ is the contact angle of water, °; $\eta$ is the water viscosity of seedling substrates; $r_1$ and $r_2$ are the contact particle radius and the water film radius at the adhesion interface, respectively, m; $t$ is the separation time of soil and non-soil material, s; $k_1$ and $k_2$ are the water film contact coefficients.

Previous studies showed that the pick-up pins at the side distance of 2-3 mm from the cell width could grasp and hold the root lump as much as possible [14]. Since the pin was made of 304 stainless steel wire with 1.6 mm in diameter, the side distances from the upper and lower pick-up pin inlets to the hole were more than 2.8 mm (the sum of the minimum side distance and the pin's radius). Moreover, the seedling's roots peripherally wrapped the rising substrate [29]. This side distance from the upper and lower pick-up pin inlets to the hole should be enlarged appropriately to decrease the picking injuries to the roots. When the height of the root lump is constant, it increases grasping penetration depth by enlarging the pick-up upright inclination angle. However, an excessively large pick-up angle would interfere with the bottom opening of the pins. Based on the specification sizes, the penetration angles along the diagonal hypotenuse were calculated at 32° and 24° for the 128-cell and the 72-cell tray, respectively. Also, it was essential to determine the corresponding pick-up penetration depth when considering the cylinder finger's stroke.

The theoretical analysis showed that the resultant force of $F_{G+A}$ was related to the seedling's gravity, the consolidation ability of the root-soil, the water content of the root lump and the adsorption effect of substrate particles. There should be a positive correlation between the resultant force and the peak force of pulling seedlings [28]. In the practical application, this approximate value for the resultant force was obtained by the mechanical test method of pulling seedlings. Taking the Hezuo 906 tomato seedlings produced in the 72-cell trays as test objects, seeds were sown in the tray cell with 22 mL of growth media and finally covered with about 2 mm of fine vermiculite. Seedling production was conducted to meet the Agricultural Professional Standard of China (General rule for vegetable plug transplant production: NY/T 2119-2012 [30]). The tomato seedlings had 38 days of growth after seeding, and the following four days of 'tempering' treatment were carried out before testing. The moisture contents of the root lumps in transplanting were set at the moderate range of 55% to 60%, which would be suitable for extracting seedlings [7,8].

As shown in Figure 5, the plug seedlings were pulled out by the universal testing machine (Accuracy level: 0.5) to measure the mechanical properties in the desorption process. There were 20 seedlings for continuously pulling in each test. The same mechanical test of pulling seedlings was repeated three times. With the increase in the pulling distance, the seedling's ability to resist extraction was significantly strengthened. When the seedling was removed from the tray cell, the associated force was approximately equal to the seedling's gravity. In the study, the maximum peak force of pulling seedlings was approximately

equivalent to the resultant force from the seedling's gravity and its adhesion force in the vertical direction.

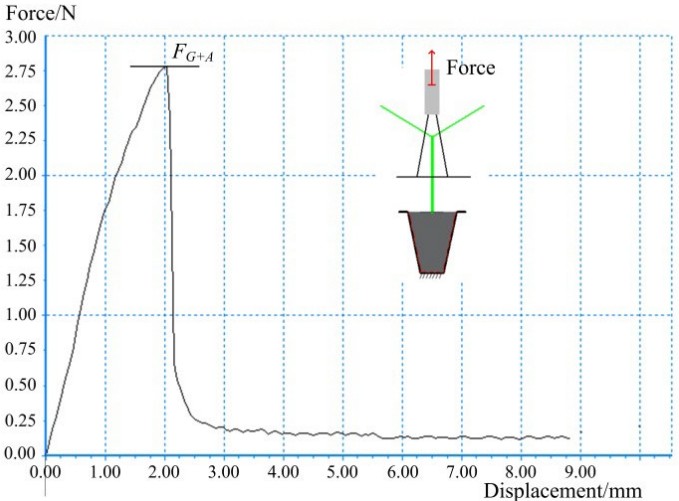

**Figure 5.** Mechanical tests of pulling seedlings under the quasistatic loading conditions.

The test results of pulling seedlings are shown in Table 1. By substituting the mean value into Equation (4), the spring tightening forces could be calculated at 0.57 N to 0.88 N according to the penetration angles for the 128-cell and the 72-cell tray cell. Given the engineering safety factor of the non-tight closing and inelastic deformation of pick-up pins, these pre-tightening forces might be appropriately magnified to stabilize the root lump grasping. According to the amount of grasping deformation, the stiffness coefficient of the tightening spring could be preliminarily determined. So, the appropriate tightening spring would be selected for the multi-pin grasping type seedling gripper.

**Table 1.** Results of seedling pulling testing and computational analysis in picking seedlings.

| Group | $F_{G+A}$ (N) | | | | $F_{S1}$ (N) | |
|---|---|---|---|---|---|---|
| | Minimum Value | Maximum Value | Mean Value | Standard Deviation | $\theta$ (128-Cell) | $\theta$ (72-Cell) |
| 1 | 1.80 | 2.75 | 2.19 | 0.24 | 0.62 | 0.87 |
| 2 | 1.55 | 2.68 | 2.21 | 0.32 | 0.63 | 0.88 |
| 3 | 1.67 | 2.82 | 2.03 | 0.28 | 0.57 | 0.81 |

### 2.2.2. Transplanting Manipulator

As shown in Figure 6a, a combination of synchronous belt linear actuators was used to create the transplanting manipulator labeled X-axis and Y-axis matrixes in the Cartesian coordinate system. Two linear actuators of the same construction were mounted side by side in the Y-axis direction, connected through a long connecting shaft. Another linear actuator was installed at the X-axis direction, which was straightly hung onto the sliders of two Y-axis linear actuators. The gripper moving horizontally and longitudinally was fixed onto the slider of the horizontal linear actuator. Each linear actuator was supported by lightweight aluminium alloy wires, and two limit travel ends were installed with the flexible rubber cushion for the sake of reducing vibration.

Moreover, two limit positioning sensors were allocated, and one start positioning sensor was used to detect the zero position. For the precise motion, each linear actuator was driven by a servo motor allocated with a high-performance servo driver. Besides, the upright direction was allocated with a self-locking commutator. After power-off, the Y-axis linear actuators could not spontaneously rise or fall.

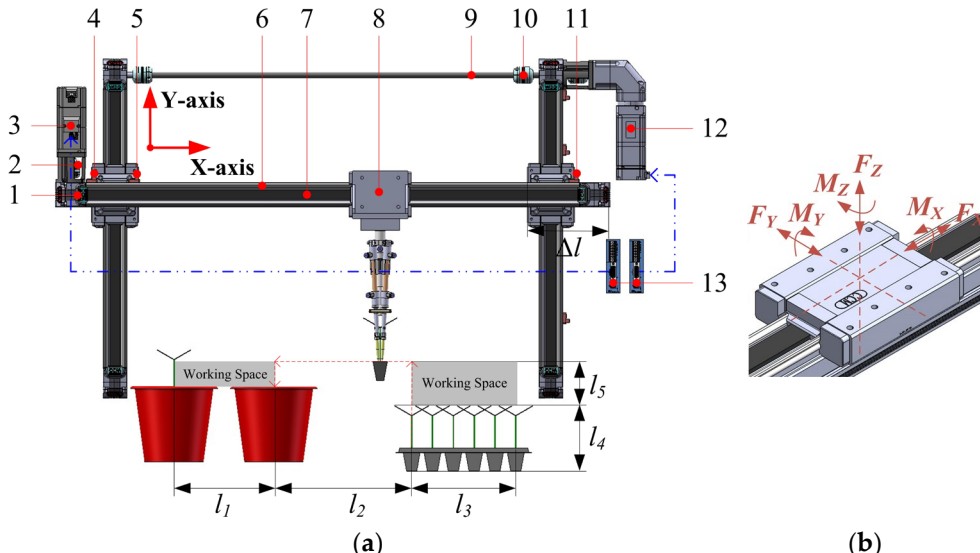

**Figure 6.** Mechanical structure drawing of the Cartesian coordinate transplanting manipulator: (**a**) overall structure: 1. rubber cushion; 2, 10. diaphragm coupler; 3, 12. servo motor; 4, 11. limit positioning sensor; 5. start positioning sensor; 6. rail beam; 7. synchronous belt; 8. slider; 9: connecting shaft; 13: servo drives; (**b**) force checking.

According to the working space of the robotic transplanting workcell, the total effective length of the transplanting manipulator should meet the following design conditions.

$$\begin{cases} L_{HD} \geq l_1 + l_2 + l_3 + 2\Delta l \\ L_{VD} \geq l_4 + l_5 + 2\Delta l \end{cases} \tag{6}$$

where $L_{HD}$ and $L_{VD}$ are the total lengths of the linear actuators in the horizontal direction and the vertical direction, respectively, mm; $l_1$, $l_2$, and $l_3$ are the center distance of the corresponding group of flowerpots, the distance from the flowerpot center to the first tray, and the largest distance of the total tray cells, respectively, mm; $l_4$ and $l_5$ are the total height of plug seedlings and the height of a flowerpot exceeding the plug seedling, respectively, mm; $\Delta l$ is the shared dimensions of the single-end installation and fixation in each actuator, mm.

To meet the high-speed operation, the selection design of the linear actuator should be carried out under the calibration load of its slider (Figure 6b). The force-checking formula was expressed as follows.

$$\lambda = \frac{F_Y}{F_{YMAX}} + \frac{F_Z}{F_{ZMAX}} + \frac{M_X}{M_{XMAX}} + \frac{M_Y}{M_{YMAX}} + \frac{M_Z}{M_{ZMAX}} \leq 1 \tag{7}$$

where $\lambda$ is the maximum load coefficient of operation life in the linear actuators; $F_Y$ and $F_Z$ are the loads of the linear actuator sliders at different directions, respectively, N; $F_{YMAX}$, $F_{ZMAX}$ are the calibrated loads of the linear actuator sliders at different directions, respectively, N; $M_X$, $M_Y$ and $M_Z$ are the load torques of the linear actuator sliders at different directions, respectively, N.m; $M_{XMAX}$, $M_{YMAX}$ and $M_{ZMAX}$ are the calibrated load torques of the linear actuator sliders at different directions, respectively, N.m.

It was assumed that the plug tray was transported along its length direction, and there were two growth pots waiting to be planted. Moreover, the outer diameter of the widely used growth pot was about 200 mm, and its vertical height was up to 300 mm. For Pepper seedlings (*Capsicum anmuum* L.), Tomato seedlings and Cucumber seedlings, the maximum plant height was close to 200 mm [8]. So, the total length of the horizontal linear actuator and the vertical linear actuator should be over 680 mm (the double pot diameter of 400 mm and the tray width of 280 mm) and 500 mm (the sum of the pot height and the seedling

height), respectively. Given the common sizes at both ends, the actual selection lengths of the linear actuators may be larger. The maximum load coefficient is the operation life coefficient of each linear actuator running by 10,000 Km, which is the basis for the load calibration of actuators. The specific design should also consider the force and moment generated by the acceleration of single-axis or multi-axis motion. In this study, the picking force of the seedling gripper was no more than 5 N, which was relatively small on the slide block of the linear actuator [8]. The inertial forces on the moving parts were also not very large. So, the actual operation parameters for regular use could all meet the requirement of the maximum load coefficient.

### 2.2.3. Parallel Conveyor

As shown in Figure 7, the double-row chain-type conveyor was designed to move these plug trays or growth pots to the working position of the seedling gripper in parallel. The two sides of the conveyor were allocated with long baffles for adjusting the bilateral distance, which could also guide the destination trays and pots to be transported in the appointed direction during seedling transplanting. On the bilateral chains, some equispaced push rods were distributed at the same size of the tray/pot lengths along the transport direction, which were used to push them forward, making the transportation without slippage. Since the chain transmission meshed like an intermediate flexible body, a tensioner at the driven sprocket supporting shaft was added to reduce vibration for cushioning and improve transportation precision. The conveyors were driven by the servo reducer motors, which could start and stop at a short distance as needed. The corresponding position sensors and emergency stop switches were configured for precise control and security protection. The photo sensors were used to detect the front edge of a plug tray or growth pot in the working position. After the entire row of seedlings was transplanted, the parallel conveyors were moved forward to place another row of seedlings or growth pots into the working location. This working procedure was repeated until all seedlings in the plug trays were transplanted.

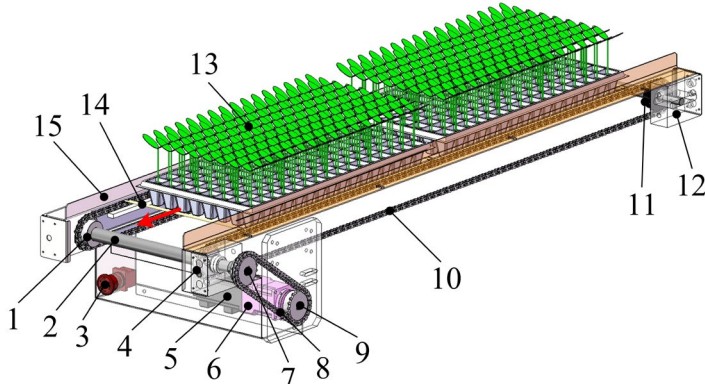

**Figure 7.** Structure drawing of the chain parallel conveyor: 1, 7, 9, 11. sprocket wheel; 2. connecting shaft; 3. emergency stop switch; 4. frame; 5. servo motor; 6. reducer; 7. synchronous belt; 8, 10. chain; 12. tensioning mechanism; 13: plug seedling; 14: pushing rod; 15. side baffle.

### 2.3. Development of Control System

Based on the planned transplanting requirements, the control system should manipulate the gripper to continuously finish a work cycle of approaching, grasping, extracting, transferring, and planting a seedling. It was also necessary to coordinate the material conveying of the source trays and the destination pots. The seedling gripper was taken as a pneumatic claw structure in use. Some five ports and two positions of solenoid control valves were used to drive these double-acting cylinders. With the help of an overflow-relieve valve, the inlet path of each cylinder was constantly adjusted to meet various working pressures. An exhaust silencer throttle valve was installed on the outlet

for different working speeds. The grippers were designed with multichannel switch status detection and on-off controls, which could respond to the operating action in time. In the simplified greenhouse robot transplanting workcell, the mature servo motor control was adopted for the reciprocated displacement of the seedling gripper and the conveying of plug trays/growth pots. The flexible S-shape acceleration and deceleration control algorithm was used to eliminate the motion error [8]. The corresponding starting and limiting locations were collocated with several sensors for positioning.

Overall, the feeding system was synchronized with the drive system of the seedling pick–up the device so that the gripper extracted the seedlings one by one. For the convenience of manipulation, a well-established PLC system was used as the host controller. A touch screen was allocated for man-machine interaction, corresponding power supply module and air supply power. The final hardware configuration structure is shown in Figure 8.

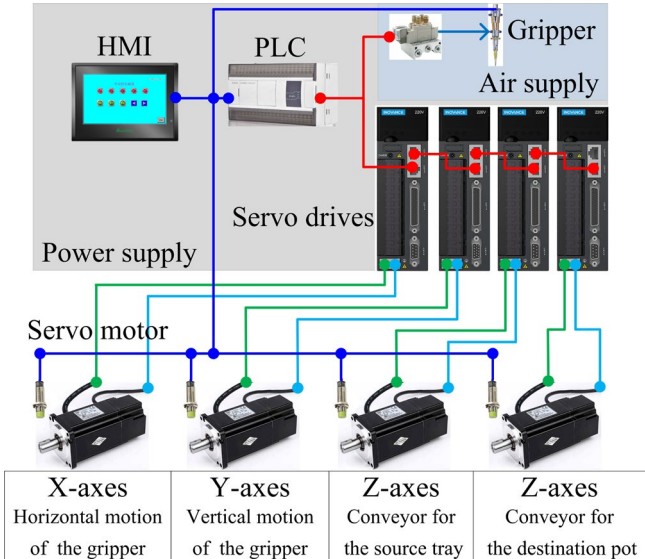

**Figure 8.** Hardware structure drawings of the control system for the automatic greenhouse robotic transplanter.

The control flow chart is illustrated in Figure 9a. After the power-on and start-up control, a self-test detection was performed to check whether the working states of the actuators and positioning sensors were normal. Before transplanting, the operating modes were set, which mainly included the working trajectories and working velocity of the grippers (Figure 9b). The working trajectories included the displacement of the column, the length and width of displacement, and the rising and falling distances in the upright direction. The horizontal motion of the seedling gripper may be programmed to be linear (straight-line) or curvilinear (joint-interpolated). When a straight-line motion is used, each linear speed may be specified. This flexible automation, through introducing a procedure to accommodate many changing parameters, might become an important alternative to traditional mechanization procedures. Finally, an order transplanting mode was generated and downloaded to the PLC. So, such a greenhouse transplanting mode was constructed on-site. The application scenario was that the entire rows of seedlings were transplanted one by one in order, and the conveyors continuously moved the plug trays/growth pots to the gripper's working space. The transplanting procedure was repeated until all seedlings in the plug trays were transplanted as needed.

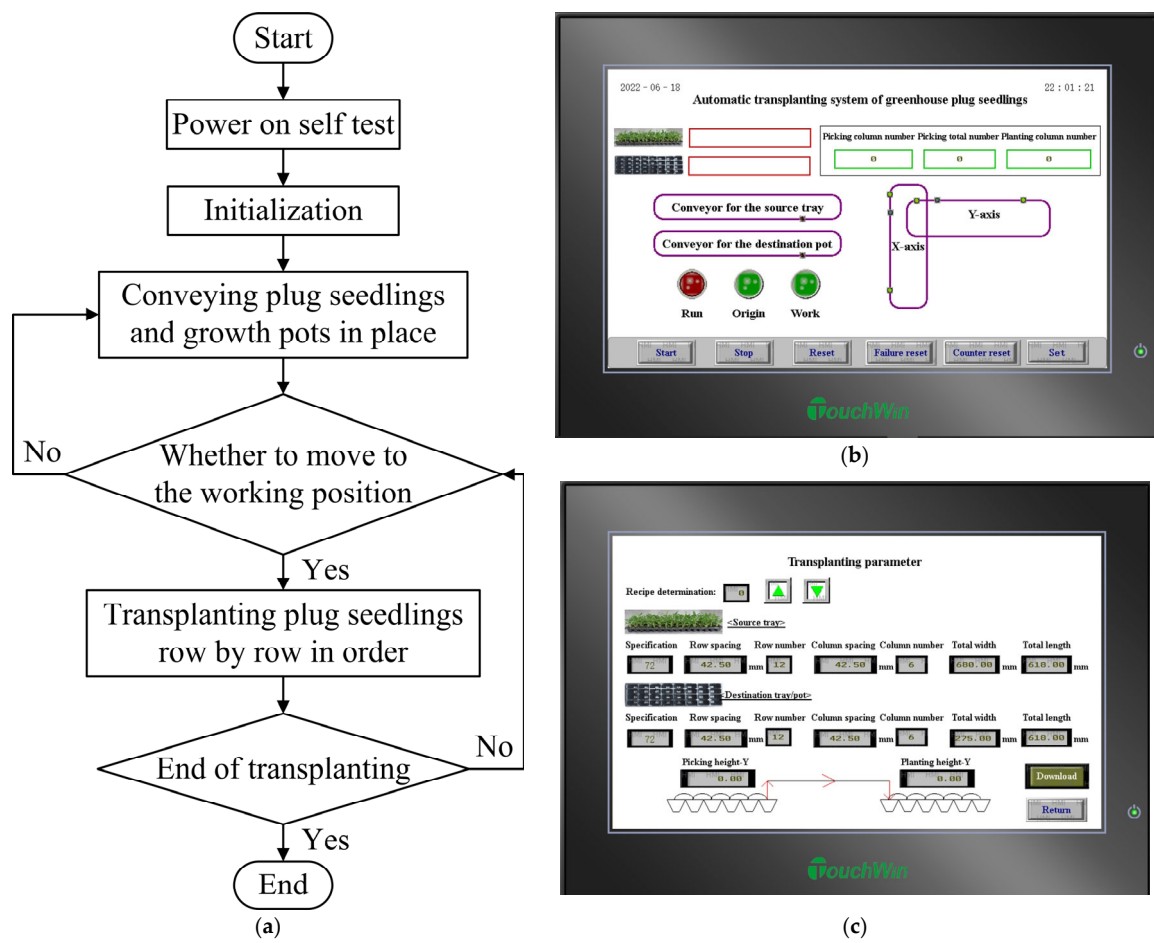

**Figure 9.** Software structure drawing of the control system and its parameter setting: (**a**) Control flow chart; (**b**) Self-test detection of sensors and other electrical components; (**c**) Initialization of working trajectory and operation parameters.

### 2.4. Construction of Physical Prototype

Using the aluminum profile as the supporting frame, the simplified greenhouse robotic transplanting workcell was constructed to examine whether or not its functional requirements were satisfied (Figure 10). The double-acting cylinder for the drive of picking pins was an MA-16 × 40-S-CA type microcylinder with an effective stroke of 40 mm from the Airtac International Group (Taiwan, China). It has an operating speed range of 30 m/s to 800 m/s and an operating air pressure range of 0.1 MPa–1.0 MPa. The open and close actions of the seedling gripper are controlled by an RBP017RCA-type rubber bag made by the Koganei Corporation, Japan. A CCM-W50-25 type belt-driven linear actuator from Dongguan Yuancheng Automation Equipment Co., Ltd., Guangdong, China, was used to construct the horizontal and vertical reciprocating movement of the seedling gripper. Its positioning repeatability is less than 0.05 mm, and the maximum load capacity is 25 kg. The used servo motor was an ISMH4 (400 W)-40B30CB type servo motor system produced by Suzhou Huichuan Technology Co., Ltd., Jiangsu, China. It has a power of 400 W, a speed response frequency of 1.2 KHz and a rated speed of 3000 rpm. The feeding system of the two-chain transmission was synchronized with the drive system of the transplanting device. So, the gripper could extract and plant the seedlings one by one. The main controller was an XD5-48T6-C type PLC controller produced by Wuxi Xinjie Electric Co., Ltd, Jiangsu, China. It has 48 input and output points, high-speed counting (up to 80 KHz), 2–10 shaft pulse output (the maximum of 100 KHz), and frequency measuring ability, and supports the X-NET field bus function. A TG-NT bus communication touch screen, an IO expanded module, several corresponding electric assisting elements (e.g., switch, power supply)

and some pneumatic elements (e.g., air compressor, solenoid valve, pressure regulating valve) were designed and equipped for the control system. According to the transplanting requirements, the sequential control software was programmed.

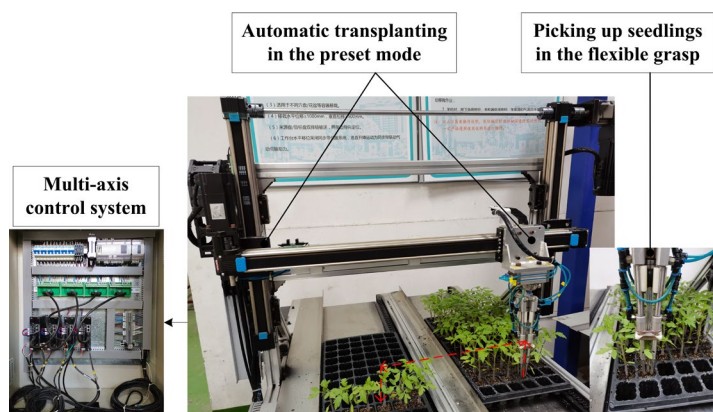

**Figure 10.** Performance tests of the simplified greenhouse robotic transplanting workcell.

### 2.5. Experiment of Multi-Factor Optimization

It was intended at the beginning of the study to concentrate maximum efforts on making the function of the robotic transplanting system more accurate and applicable. Under the standard seedling-raising conditions, the growth of plug seedlings is relatively uniform. So, the mechanical factors have a significant effect on the percentage of successfully transplanting seedlings from their growth cells [8,14]. This paper will test and evaluate the adaptability of relevant mechanical operations in the Key Laboratory of Modern Agriculture Equipment and Technology, Ministry of Education, Jiangsu University, Zhenjiang, China. The test objects were the conventional cultivated tomato pothole seedlings used in the mechanism design. According to the previous design, the pick-up pins were made of 304 stainless steel wire with a diameter of 1.6 mm [8]. Their spacing was kept at 3 mm less than the cell wall width as slanting into the root soil of the seedling. The gripper grasps the root system to lift the seedling in a vertical direction out of the cell. So, this lifting motion would affect the extraction rate, which was closely related to the transplanting efficiency. Thus, the slow, medium, and high speeds were set for transplanting operation at which the transplanting frequencies were 15 plants/min, 20 plants/min and 25 plants/min from the source tray to the growth pot. The spring tightening force of the pick-up pins was theoretically deduced in design. The grasping forces at 0.9 N, 1.2 N, and 1.5 N were tested to compare seedling extraction. According to the 72-cell dimension, the maximum penetration depth was 45 mm. The penetration depth with minus 2 mm coverage was divided into three levels of 30 mm, 35 mm, and 40 mm. As shown in Table 2, the multifactor orthogonal tests of several factors and levels were conducted according to the above design analysis.

**Table 2.** Factors and levels of the orthogonal experimental design.

| Factors (Levels) | A: Transplanting Frequency (plant/min) | B: Grasping Force (N) | C: Penetration Depth (mm) |
|---|---|---|---|
| 1 | 15 | 0.9 | 30 |
| 2 | 20 | 1.2 | 35 |
| 3 | 25 | 1.5 | 40 |

Low damage to the seedling's root lump during the transplanting process is the key to ensuring growth and development in the later stage (7,8,14). So, the integrity rate of root lumps in automatic transplanting was chosen as the index of transplanting experiments. It was defined as follows.

$$IR_{AT} = [W_1/(W_1 + W_2)] \times 100 \tag{8}$$

where $IR_{AT}$ is the integrity rate of automatic transplanting, %; $W_1$ is the residual weight of a plug seedling after transplanting, g; $W_2$ is the damage weight during automatic transplanting and dropping, g.

To facilitate the statistics of the root lump damage, plug seedlings were transplanted one by one into the same plug trays, as shown in Supplementary Video S1. Based on these factors and levels, the $L_9$ ($3^4$) type orthogonal table was used for the multi-factor optimization. There were nine testing groups, and the unused column was used as the error. In each test, the whole-tray seedlings were continuously transplanted, and the process was repeated three times. The corresponding results were recorded on-site, and the statistical analysis was conducted using the SPSS 18.0 software package (SPSS Inc., Chicago, IL, USA). Analysis of variance (ANOVA) was carried out with the least significant difference method (LSD).

### 2.6. Experiment of Transplanting Performance

The transplanting performance tests were used to further check the optimal operation parameters. The test objects were typical vegetable seedlings such as tomato seedlings, cucumber seedlings and pepper seedlings. The used seedlings for transplanting were produced in 128-cell and 72-cell trays by the local farmers. Their growth characteristics are presented in Table 3. In the test trials, these seedlings from the 128-cell and 72-cell trays were transplanted into the 50-cell trays. Each test was repeated five times with a one-month interval. The success ratio in transplanting seedlings represents how successfully the device performs extracting, transferring, and discharging of seedlings. Besides, pick-up failures, discharging failures and breakage of root lumps were considered special functional failures. The corresponding results were recorded, and the percentage analyses were conducted on the raw data.

**Table 3.** Seedling growth characteristics in transplanting performance tests.

| Tray | Seedling | Seedling Age (day) | No. of Leaves | Seedling Height (mm) | Leaf Length (mm) | Leaf Width (mm) |
|------|----------|--------------------|---------------|----------------------|------------------|-----------------|
| 128 | Tomato | 32 | 4~5 | 115.20 ± 5.75 | 39.15 ± 6.94 | 24.18 ± 4.53 |
| | Cucumber | 20 | 2~3 | 75.26 ± 6.15 | 42.23 ± 4.18 | 23.96 ± 5.86 |
| | Pepper | 47 | 5~6 | 174.58 ± 9.25 | 34.19 ± 5.87 | 23.26 ± 4.14 |
| 72 | Tomato | 41 | 5~6 | 131.65 ± 5.28 | 39.43 ± 4.67 | 24.09 ± 4.63 |
| | Cucumber | 31 | 3~4 | 87.46 ± 6.15 | 62.54 ± 6.43 | 35.68 ± 6.44 |
| | Pepper | 54 | 6~7 | 182.17 ± 6.41 | 37.19 ± 5.75 | 28.62 ± 4.17 |

Note: Data is mean ± std. dev.

### 3. Results

#### 3.1. Multi-Factor Optimization Experiment

Seedling transplantation would provide a critical jump in the production season and mitigate risk by growing in a controlled environment for germination and young-plant stages. The multi-factor optimization experiment's results are shown in Table 4. The maximum range R indicates that the corresponding factor has a great effect on the evaluation index. Viewed from the range comparison, the range R of the transplanting frequency and pick-up penetration depth were separated into the first and second factors affecting the integrity rates of seedling transplanting. In the designed test conditions, the grasping force had little effect on the success of seedling extraction. The optimal level group was A1B2C2. When the transplanting frequency was 15 plants/min, the tightened spring force was 1.2 N, and the pick-up penetration depth was 35 mm, the optimum effects of automatic transplanting seedlings could be achieved at last. The average integrity rate of root lumps was above 90%, which could ensure the integrity of the seedlings in the transplanting process.

**Table 4.** Results of the multi-factor optimization experiment.

| Test Number | Factor (Level) | | | | Integrity Rate (%) |
|---|---|---|---|---|---|
| | **A (3)** | **B (3)** | **C (3)** | **Dᵃ (3)** | |
| 1 | 1 (15 plant/min) | 1 (0.9 N) | 1 (30 mm) | 1 | 90.79 |
| 2 | 1 | 2 (1.2 N) | 2 (35 mm) | 2 | 93.63 |
| 3 | 1 | 3 (1.5 N) | 3 (40 mm) | 3 | 92.11 |
| 4 | 2 (20 plant/min) | 1 | 2 | 3 | 90.57 |
| 5 | 2 | 2 | 3 | 1 | 90.24 |
| 6 | 2 | 3 | 1 | 2 | 88.36 |
| 7 | 3 (25 plant/min) | 1 | 3 | 2 | 88.75 |
| 8 | 3 | 2 | 1 | 3 | 88.46 |
| 9 | 3 | 3 | 2 | 1 | 89.21 |
| $K_1$ | 92.18 | 90.04 | 89.20 | | |
| $K_2$ | 89.72 | 90.78 | 91.14 | | |
| $K_3$ | 88.81 | 89.89 | 90.37 | | |
| R | 3.37 | 0.88 | 1.93 | | |
| Optimal level | A1 | B2 | C2 | | |

Note: Dᵃ is Error.

The statistical analysis of variance (ANOVA) was shown in Table 5. The *p*-value result showed that the transplanting frequency had a highly significant effect on the integrity rate of seedling transplanting, and the penetration depth had a significant effect. For the grasping force, there were no significant effects on the integrity rate of seedling transplanting ($p > 0.05$). The results obtained from the statistical analysis of variance were consistent with the orthogonal tests. Overall, the success or failure of automatic transplanting largely depended on whether the manipulator could reliably grasp plug seedlings.

**Table 5.** Analysis of variance (ANOVA) for the multi-factor optimization experiment.

| Source | Sum | DOF | Mean Square | F-Value | *p*-Value | Significance |
|---|---|---|---|---|---|---|
| A: Transplanting frequency | 18.216 | 2 | 9.1080 | 134.3805 | 0.0074 | ** |
| B: Grasping force | 1.3484 | 2 | 0.6742 | 9.9474 | 0.0913 | ns |
| C: Penetration depth | 5.6840 | 2 | 2.8420 | 41.9313 | 0.0233 | * |
| D: Error * | 0.1356 | 2 | 0.0678 | | | |
| Sum | 25.384 | | | | | |

Note: ns, no significant effect; *, ** significant level at $0.01 < p < 0.05$, $p < 0.01$, respectively.

### 3.2. Transplanting Performance Experiment

Under the optimal combination parameters, the transplanting performance of the robotic transplanting workcell was further evaluated in a laboratory. Taking tomato, cucumber and pepper seedlings as the transplanting objects, various seedlings were transplanted into the 50-cell growing trays. The transplanting process was observed and recorded on-site using a CCD camera. The corresponding results are shown in Table 6. The maximum success ratio for automatic transplanting was 95.47% for 128-cell trays of tomato seedlings, and the minimum success ratio was 91.67% for 128-cell trays of pepper seedlings. In general, the success ratio of pick-up seedlings at 128-cell trays was higher than that at 72-cell trays. The successful possibilities of transplanting seedlings increased as the tray cells were relatively small. The reason might be that these small tray cells tend to produce sturdy seedling plugs with well-developed roots.

**Table 6.** Results of the automatic transplanting performance experiment.

| Plug Tray | Seedling | No. of Seedlings Fed | No. of Pick-Up Failures | No. of Soil Lump Breakage | No. of Seedling Damages | No. of Discharge Failure | Success Ratio % |
|---|---|---|---|---|---|---|---|
| 128-cell | Tomato | 640 | 12 | 9 | 3 | 5 | 95.47 |
| | Cucumber | 640 | 11 | 6 | 16 | 7 | 93.75 |
| | Pepper | 640 | 8 | 7 | 13 | 3 | 95.16 |
| 72-cell | Tomato | 360 | 7 | 5 | 7 | 3 | 93.89 |
| | Cucumber | 360 | 4 | 7 | 12 | 4 | 92.50 |
| | Pepper | 360 | 8 | 6 | 10 | 6 | 91.67 |

## 4. Discussion

In the face of relatively uniform living objects, the key mechanical operations involved in the study had different transplanting effects. With the increase in working frequency, the automatic transplanting quality of plug seedlings decreased. The average integrity rate of the root lump was up to 92.18% at the low-speed steady operating condition of 15 plants/min, which was higher than the other two transplanting frequencies. Increasing inertial impact was associated with interweaving the gripper's grasping, extracting, transferring, and discharging actions on the root lump, possibly making the pick-up pins break through the root-soil complex ceaselessly. If the seedling's root lump was sturdy enough with the large root volume in the tray cell, it could withstand the toss of fluctuating transplanting from side to side. This requirement was also essential for tolerance of transport to the field, the trauma of rapid handling on the machines, and survival after pot or field planting. Therefore, high-quality seedling production was necessary as the full potential of automatic transplanting was to be realized [8,18]. Adding reinforcement material was an effective means to improve seedling quality [31]. In future research, the flexible grasping design should be considered so that the gripper can widely adapt to the characteristics of plug seedlings.

The multi-pin seedling pick-up gripper was designed to effectively grasp, hold, and release plug seedlings [8]. Under the conditions of three grasping forces, the integrity rates of root lumps were not significantly different. It showed that this kind of multi-pin centripetal grasping could adapt well to the root-soil structure of plug seedlings. In the trials, the success ratio of the maximum grasping action was superior to that of the two weak powers. It was consistent with previous studies, which emphasized that the picking pins must hold seedlings firmly [14]. Once a seedling was successfully grasped from the tray cell, there was the possibility of a complete transplantation into the growth pot. Since the closing motion of the designed gripper mainly depended on the tightening action of the flexible spring, the required grasping forces on the root lump could be obtained by replacing the looped spring with different stiffness or lengths. It is well known that fatigue damage is one of the key factors in the failure of compression springs. In application, the reliability of tightening grasp should be checked regularly.

Since these plug seedlings grow inherently in the narrow tray cells, it is not easy to pull them out. In many cases, the seedling stems are fragile by nature. The best way to extract is to excavate the root-soil as deep as possible. For the pincette-type pick-up methods, it is particularly critical for the gripper to grasp the maximum amount of root lump in the tray cell [8,14]. When the penetration depth of the picking pins was up to the maximum value of 40 mm, the seedling integrity in automatic transplantation was more than 90%. This was superior to another two pick-up depths. Limited by the narrow spacing of the tray cell, the penetration depth was always out of reach. If the penetration depth was too shallow, it might be insufficient for the pick-up pins to grasp the seedling. When plug seedlings were not well coiled by their roots, the shaped root lump often partly scattered during the extraction [18,32]. The seedlings could be extracted successfully as the pick-up depth was up to 3/4 cell depth. This penetration depth control would be achieved by appropriately adjusting the lifting height of the gripper.

In automatic transplanting performance tests, the percentage analysis was made between the unsuccessful transplanting plants and the integral fed seedlings (Figure 11). The pick-up failures and the root-soil breakage were less than 5%, which showed that the designed transplanting system was satisfied with grasping seedlings from the tray cells and transferring them to the growth pots for planting (Figure 11a,b). For three common vegetable seedlings, pick-up failures sometimes occurred. Compared with tomato and pepper seedlings, there was less possibility of failure in picking cucumber seedlings. For the 128-cell and 72-cell trays of tomato seedlings, the pick-up failures and the root-soil breakage were similar. Transplanting failures easily occurred when pepper seedlings were grown in large trays. A plunger was used to push out of the unsuccessful plug seedlings. Since the root soils were not well developed, most transplanting failures often occurred with those weak seedlings [7,18]. In general, plug seedlings should have a well-developed root system [33]. At the same time, the roots must be evenly distributed in the rhizosphere soil so that the growth media is not broken during transplantation [26,27]. Once transplanted, they should tolerate mechanical challenges and continue growing to achieve optimum yield. This is consistent with previous findings that successful extraction of plug seedlings depends on the root coiling state [8,14].

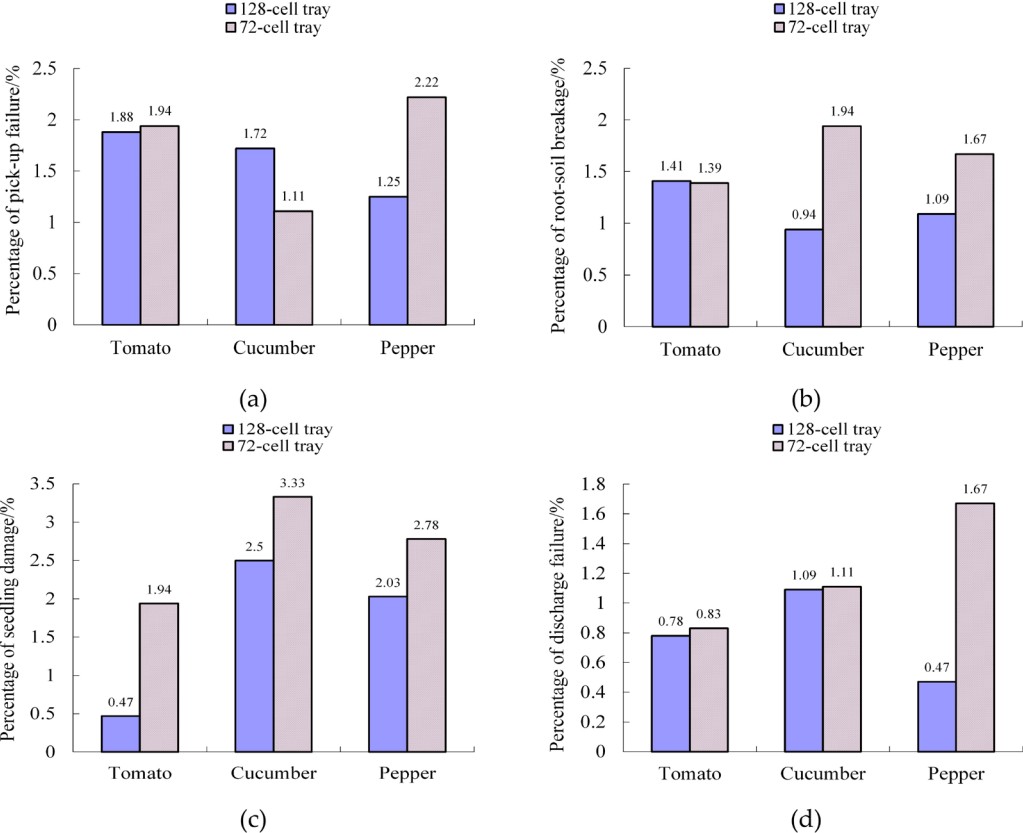

**Figure 11.** Failure analysis in automatic pick-up seedlings from the tray cells: (**a**) pick-up failure; (**b**) root-soil breakage; (**c**) seedling damage; (**d**) discharge failure.

For the injuries of seedling plants, stem breakage was severe in the cucumber and pepper seedlings (Figure 11c). Since 72-cell trays of pepper seedlings were relatively upright, their plants were broken in the picking and discharging. The situation was mainly that the seedling's leaves would snag on the tips of the fingers as the gripper moved down toward the base of a seedling, which would cause the seedling to bend or buckle. In comparison, the tomato plug seedlings were so hard that no excessive injuries occurred during transplanting. As we all know, cucumber seedlings have wide branches and leaves, and pepper seedlings are tall and upright. Especially, those seedlings were grown in

72-cell trays, which were often tangled with the working gripper in transplanting. When the gripper discharged cucumber and pepper seedlings into the destination pots, it was difficult to discharge them with precision. The discharge failure possibility of cucumber and pepper was higher than that of tomato (Figure 11d). There was consequently a need for a vegetable-specific study on seedling qualities for compatibility with mechanized transplanting operations [33,34]. Short seedlings might reduce the problem of entanglement when the gripper moves along a straight-down approach path to the seedlings in the plug [28,34]. It was also necessary to further optimize the working parts in the direct contact and their motion trajectory. For this purpose, the control program could be modified so that the transplanting manipulator approached the seedlings using a vertical prestart of shifting movement between the plants [35,36]. Considering seedlings to be living and flexible, the synergistic innovation from horticultural and engineering perspectives should be further strengthened in terms of success in transplanting seedlings [7,16,32,34].

Healthy transplants could initiate the process of successful crop production. When performed in accordance with good, mechanized practices, these main vegetables would grow evenly and mature at similar stages, increasing the likelihood of rejection by consumers and markets [3]. The performance tests showed that the average success ratio in transplanting seedlings for the three local vegetable crops was more than 90% when the transplanting rate was 15 seedlings per minute. The overall transplanting effect reached the expected goal, which could realize the efficient and low-cost replacement of manual transplanting production. This sophisticated equipment would also be operated day and night. The automatic transplanting workcell was constructed with conventional linear modules, chain drives and cylinder assemblies. The flexible pneumatic gripper was proved to effectively penetrate, grasp, hold, and release various root lumps with minimum damage to the seedlings. This prompted vegetable production to reach less waste of seedling resources at the farm level. In future, an effort will be made to construct a multi-task robotic transplanting system to achieve more efficient automatic pipeline transplanting of greenhouse seedlings. To further reduce drudgery and the labor force associated with the transplanting process, the filling unit should bring the soil up to the filling head, push the soil into the pots, and dibble holes. For these extreme production conditions, the robotic transplanting stem would be studied to handle failures and adapt to various seedlings. It might usually require significant levels of autonomy-supported sensory feedback [11]. The development of such an integrated system would be a logical continuation of this robotic plug transplanting research. For the effects of different morphological and physical characteristics of plug seedlings, the adaptability to fully automatic machinery might be investigated for high-quality transplants [31]. The development of seedling low-loss transplantation with the coordinated promotion of minimally processed vegetables would effectively improve the utilization level of fresh vegetables [3,12]. Through continuous technological innovation, it is conducive to the healthy development of the vegetable industry while protecting the interests of upstream vegetable farmers.

## 5. Conclusions

Based on actual seedling growth and development requirements, a simplified robotic transplanting workcell was designed and evaluated for the high quality and efficiency of seedling planting. Its overall structure of a simple gantry structure robotic mechanism was designed with the X-Y Cartesian coordinate system. The operational end-effector was penetrating, holding, and extracting type pneumatic flexible gripper using four elastic pick-up pins pulled by the cylinder fingers. A set of multi-axis motion control and multi-sensor detection systems was designed to automate the process of seedling transplanting. Under the standard seedling raising conditions, the performance tests were conducted to determine the optimal operation parameters and transplant production conditions. The testing results showed that the key mechanical operations had different transplanting effects. With the increase in working frequency, the automatic transplanting quality of plug seedlings decreased. Increasing inertial impact of the gripper's grasping, extracting,

transferring, and discharging actions on the root lump might make the pick-up pins break through the root-soil complex. The seedlings could be extracted successfully as the pick-up depth was up to 3/4 cell depth. The maximum success in transplanting seedlings was 95.47% for local vegetable crops. For a stable vegetable production and marketing chain, the development of seedling low-loss transplantation should be further studied with the coordinated promotion of minimally processed vegetables.

**Supplementary Materials:** The following supporting information can be downloaded at https://www.mdpi.com/article/10.3390/app131810022/s1, Video S1: transplanting production of greenhouse seedlings.

**Author Contributions:** Conceptualization, L.H. and H.M.; methodology and design, D.X.; validation, X.D. and Q.X.; formal analysis, D.X.; investigation, G.M.; data curation, X.D.; writing—original draft preparation, D.X., Q.X. and D.X.; writing—review and editing, X.D.; supervision, L.H.; project administration, H.M.; funding acquisition, L.H. and H.M. All authors have read and agreed to the published version of the manuscript.

**Funding:** This research was funded by the National Natural Science Foundation of China (No. 51975258); the China Agriculture Research System (CARS-23-D03); the Precision and Efficient Transplanting Equipment Industrialization Demonstration Application Project (No. TC210H02X); the Open Fund of High-tech Key Laboratory of Agricultural Equipment and Intelligentization of Jiangsu Province (No. JNZ201910).

**Institutional Review Board Statement:** Not applicable.

**Informed Consent Statement:** Not applicable.

**Data Availability Statement:** All relevant data presented in the article are kept at the request of the institution and are, therefore, not available online. However, all data used in this manuscript are available from the corresponding authors.

**Acknowledgments:** The authors would like to thank the key laboratory of agricultural engineering at Jiangsu University for supporting the experimental conditions of this research.

**Conflicts of Interest:** The authors declare no conflict of interest.

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
