# Peer review of "Development of Simplified Seedling Transplanting Device for Supporting Efficient Production of Vegetable Raw Materials"

_applsci, doi:10.3390/app131810022_

Round 1
Reviewer 1 Report
The papers is well presented and has good data of novelity. It has good potential of commercial purpose. #
1-However, numerical outcomes should be incorporated in abstract.
2-The introduction is fine but objectives and hypothesis of issue/rationale of study needs to be improved.
3-Methodology is fine
4-Results section needs more elaboration of key findings.
5-Discussion needs bit improvement of data and highlighting the key outcomes of the research and its future suggestions to do more in future on this topic.
6-Conclusion should be brief and precise.
7-References: Cross check these in text and reference body and foramte as per journal style.
Author Response
Response to Reviewer 1 Comments
Point 1: However, numerical outcomes should be incorporated in abstract.
Response 1: Following the reviewer's advice, the revised manuscript has been updated with the corresponding abstract. Some important numerical outcomes have been incorporated in abstract. Thanks to the reviewer for his deep thinking.
Point 2: The introduction is fine but objectives and hypothesis of issue/rationale of study needs to be improved.
Response 2: Thanks to the reviewer for his professional advice. The relevant content has been improved in the revised manuscript, which makes the introduction more reasonable and meaningful.
Point 3: Methodology is fine
Response 3: Thanks for the reviewer's recognition. The key research points are described as fully as possible in the methodology.
Point 4: Results section needs more elaboration of key findings.
Response 4: Thanks to the reviewer for his professional advice. The results in the revised manuscript have been reorganized. The elaboration of some key findings has been condensed and edited again.
Point 5: Discussion needs bit improvement of data and highlighting the key outcomes of the research and its future suggestions to do more in future on this topic.
Response 5: Thanks to the reviewer for his professional advice. The discussion in the revised manuscript has been reorganized. The key outcomes of the research and its future suggestions have been accentuated for ease of understanding.
Point 6: Conclusion should be brief and precise.
Response 6: Thanks to the reviewer for his professional advice. The conclusion in the revised manuscript has been reorganized. The relevant research conclusion has been accentuated for ease of understanding.
Point 7: References: Cross check these in text and reference body and format as per journal style.
Response 7: Thanks to the reviewer for his meticulous review. If the manuscript is accepted for publication, we will carefully check the textual representations and references according to the format requirements of the journal.

Reviewer 2 Report
Dear Author,
Some corrections and comments are shown on the annotated manuscript. The evaluated manuscript has been uploaded to the system. The article can be published after revision according to the corrections and comments shown in the article.
Kind regards

Author Response
Response to Reviewer 2 Comments
Point 1: Young plants: replace with "seedlings"
Response 1: Following the reviewer's advice, the words of young plants have been replaced with the word of seedlings in the revised manuscript. It's easier to understand. Thanks to the reviewer for his meticulous review.
Point 2: Cucumber (Cucumis sativus L.) and tomato (Solamum lycopersicum L.) seedlings. It is not Italik?
Response 2: The words of Cucumis sativus L. and Solamum lycopersicum L. are the scientific names in Latin. If such a format is not needed, we can consider removing them in the revised manuscript.
Point 3: Figure 9. Flowchart and images have low resolution. It should be upgraded.
Response 3: Following the reviewer's advice, the flowchart and images have been made for clear display. Thanks to the reviewer for his meticulous review.
Point 4: Table 3. Plant height (mm): replace with seedling
Response 4: Following the reviewer's advice, the word of plant has been replaced with the word of seedling in the revised manuscript. It's easier to understand. Thanks to the reviewer for his meticulous review.
Point 5: Two words of seedling: remove.
Response 5: Following the reviewer's advice, the corresponding word of seedling has been removed in the revised manuscript. It's easier to understand. Thanks to the reviewer for his meticulous review.
Point 6: Table 6. No. of soil: replace with soil lump.
Response 6: Following the reviewer's advice, the word of soil has been replaced with the words of soil lump in the revised manuscript. It's easier to understand. Thanks to the reviewer for his meticulous review.
Point 7: The soil: growth media.
Response 7: Thanks to the reviewer for his meticulous review. Here, the soil is the growth media. So the corresponding word has been replaced in the revised manuscript. It's easier to understand.
Point 8: What do you think about leaf blade shape of vegetable species? Tomato: compound leaf shape. Cucumber and pepper: Simple leaf shape?
Response 8: Thanks to the reviewer for his professional advice. As we all know, cucumber seedlings have wide branches and leaves, and pepper seedlings are tall and upright. Especially, those seedlings were grown in 72-cell trays, which were often tangled with the working gripper in transplanting. Compared to cucumber and pepper seedlings, tomato plants were better.
Tomato seedlings |
Cucumber seedlings |
Pepper seedlings |
Point 9: A discussion section should be developed about the applicability of the developed prototype transplanter in different production systems (soilles culture etc) and species (leafy vegetables: lettuce, cabbage, etc.).
Response 9: Thanks to the reviewer for his professional advice. In this manuscript, we mainly develop the prototype from the perspective of applied engineering in agriculture. It was intended at the beginning of the study to concentrate maximum efforts on making the function of the robotic transplanting system more accurate and applicable. In the next step, we will carry out more extensive testing as suggested by the reviewers.
Point 10: As can be understood from the article and the video, it is seen that the planting machine developed may be suitable for the seedlings of varieties that do not grow rosettes such as tomatoes, peppers and cucumbers. The machine can transfer the seedlings of the mentioned species from the tray to another medium with 95% accuracy. However, there is no information and discussion about fixing to the planting site (soil, pot, rockwool package). This issue should be considered in the discussion section of the article.
Response 10: Thanks to the reviewer for his professional advice. Taking tomato, cucumber and pepper seedlings as the transplanting objects, the relevant tests were carried out in this paper. For the effects of different morphological and physical characteristics of plug seedlings, the adaptability to the fully automatic machine might be investigated for high-quality transplants in future. The synergistic innovation both from horticultural and engineering perspectives should be still further strengthened in terms of success in transplanting seedlings. In the next step, we will carry out more extensive testing as suggested by the reviewers.

Reviewer 3 Report
Kindly see attachment

Author Response
Response to Reviewer 3 Comments
Point 1: Abstract: Kindly check minor spelling mistake. The abstract needs a better composition of words.
Response 1: Following the reviewer's advice, the revised manuscript has been updated with the corresponding abstract. Some spelling mistakes in the abstract have been corrected. And the better composition of words has also been reorganized.
Point 2: Introduction: Authors should explain why they chose multi-pin grasping type in this work.
Response 2: Thanks to the reviewer for his professional advice. Previous studies have shown that it is good for picking up seedlings using the multi-pin grasping type operation. So the revised version of the paper makes such an explanation.
Point 3: Material and methods: Insert the area studied in this experiment if possible.
Response 3: Following the reviewer's advice, the revised manuscript has been updated with the corresponding methodology. The area studied in this experiment has been added in the text.
Point 4: Results: Results are well indicated.
Response 4: Thanks for the reviewer's recognition.
Point 5: Discussion: Could be more detailed. The lines should be more focused for highlighting the important finding of this work.
Response 5: Thanks to the reviewer for his professional advice. The discussion in the revised manuscript has been reorganized. The important finding of this work has been accentuated for ease of understanding.
Point 6: References: Strengthen the part of the discussion with 2 or 3 new references.
Response 6: Following the reviewer's advice, the revised manuscript has been updated with the corresponding references. In recent years, there have been many studies on the field transplanters. But there is relatively a little transplanting technology research in greenhouse production. We try to find some appropriate new references to cite them.
Point 7: Conclusion: Minor language issues must be addressed to improve quality of the MS.
Response 7: Following the reviewer's advice, the quality of the MS in the conclusion has been improved for ease of understanding. Thanks to the reviewer for his meticulous review.
